# A polarized cell system amenable to subcellular resolution imaging of influenza virus infection

Jean-Baptiste Brault[1]*, Catherine Thouvenot[2], Magda Cannata Serio[1¤],
Sylvain Paisant[1], Julien Fernandes[3], David Gény[4], Lydia Danglot[4,5], Adeline Mallet[2],
Nadia Naffakh[1]*

1 Institut Pasteur, Université Paris Cité, CNRS UMR 3569, RNA Biology of Influenza Viruses, Paris, France,
2 Institut Pasteur, Université Paris Cité, C2RT, Ultrastructural BioImaging Unit, Paris, France, 3 Institut
Pasteur, Université Paris Cité, C2RT, Unit of Technology and Services Photonic BioImaging, Paris, France,
4 Institute of Psychiatry and Neuroscience of Paris (IPNP), INSERM U1266, NeurImag Facility, Université
Paris Cité, Paris, France, 5 Institute of Psychiatry and Neuroscience of Paris (IPNP), INSERM U1266,
Membrane Traffic in Healthy and Diseased Brain team, Paris, France

¤ Current address: Institut Curie, PSL Research University, CNRS UMR144, Paris, France
* jean-baptiste.brault@pasteur.fr (J-BB); nadia.naffakh@pasteur.fr (NN)

Recherche en Cancerologie de Lyon, FRANCE

**Data Availability Statement:** All relevant data are
within the paper and its Supporting Information
files.

## Abstract

The life cycle of influenza A viruses (IAV), and notably intracellular trafficking of the viral
genome, depends on multiple interactions with the cellular cytoskeleton and endomem-
brane system. A limitation of the conventional cellular models used for mechanistic study
and subcellular imaging of IAV infection is that they are cultured in two dimensions (2D)
under non-polarizing conditions, and therefore they do not recapitulate the intracellular orga-
nization of the polarized respiratory epithelial cells naturally targeted by IAVs. To overcome
this limitation, we developed an IAV-infection assay in a 3D cell culture system which allows
imaging along the baso-lateral axis of polarized cells, with subcellular resolution. Here we
describe a protocol to grow polarized monolayers of Caco2-TC7 cells on static Cytodex-3
microcarrier beads, infect them with IAV, and subsequently perform immunostaining and
confocal imaging, or electron microscopy, on polarized IAV-infected cells. This method can
be extended to other pathogens that infect human polarized epithelial cells.

## Introduction

Influenza viruses cause annual winter epidemics and are the leading cause of severe respiratory
tract infections worldwide, with an estimated 500,000 deaths per year. Influenza A viruses
(IAVs), which have an animal reservoir, can also cause pandemics with potentially devastating
consequences in terms of mortality and economic loss [1]. Highly pathogenic avian IAVs of
the H5N1 subtype, that are currently endemic in most parts of the world and have been found
to infect and cause disease in many mammal species, are of great concern [2]. To support the
much-needed development of advanced vaccines and anti-viral drugs, an improved under-
standing of the molecular mechanisms involved in the viral life cycle is required.

**Funding:** NN: HFSP-RGP0040/2019 https://www.hfsp.org/ NN and AM: ANR-10-LABX-62-IBEID https://anr.fr/ NN and LD: ANR-21-CE11-0010-03 https://anr.fr/ AM: ANR-10-INSB-04-01 and ANR-10–INBS-04 LD: ANR-10-INBS-04 https://anr.fr/ The funders did not and will not have a role in study design, data collection and analysis, decision to publish, or preparation of the manuscript."

**Competing interests:** The authors have declared that no competing interests exist.

A limitation of most available data on influenza life cycle is that they derive from studies carried out in immortalized human cell lines cultured in two dimensions (2D) under non-polarizing conditions. These cellular models do not resemble the natural target tissue of IAVs, which is the polarized respiratory epithelium. Polarized and non-polarized epithelial cells exhibit a very different if not opposite organization of the cytoskeleton and molecular motors (kinesins versus dynein) used for anterograde and retrograde intracellular transport. Indeed, in non-polarized cells microtubules grow from a central centrosome towards the cell periphery, whereas in polarized epithelial cells the centrosome is apically-located and microtubules grow in the opposite direction, towards the centre of the cell (**Fig 1A**, [3]). Consistently, post-Golgi secretion of anterograde cargoes was reported to be kinesin-driven in non-polarized HeLa cells [4] while it is regulated by dynein in polarized epithelial cells [5]. Therefore, cell polarization certainly has a strong impact on the cellular factors that contribute to the intracellular transport—including entry and exit—of viral components. Human primary nasal or tracheo-bronchial epithelial cells grown at the air–liquid interface (ALI) and differentiated into ciliated, secretory and basal cells, are being increasingly used to study the innate immune responses to infection or to screen antiviral molecules [6,7]. However, ALI cultures pose severe technical limitations for mechanistic investigations, in particular when subcellular imaging is required. Indeed, the polyester porous membrane on which the cells are grown generates high levels of light scattering, and the sample thickness (several tens of microns) induces chromatic aberrations which preclude confocal imaging along the baso-apical axis with a subcellular resolution. Sectioning of ALI samples is possible, however due to the stiffness of the polyester membrane it requires paraffin-embedding [8], which leads to significant fluorescent signal quenching and therefore restricts the possibilities of highly resolved imaging at the subcellular scale.

To provide a system of human polarized cells amenable to immunostaining and confocal imaging of IAV-infected cells with a subcellular resolution, we have adapted a 3D cell culture system on static microcarriers previously described for Caco-2 and Madin-Darbin canine kidney (MDCK-II) cells, and used to study enterovirus and calicivirus infection [9–11]. We first tested various human respiratory and/or epithelial cell lines known to support IAV growth (A549, Calu-3, NCI-H292, RPMI-1, 16HBE14, Caco-2/TC7), as well as quasi-primary, BMI-1-expressing bronchial epithelial cells derived from two distinct patients ([12], kindly provided by Ian Sayers, Nottingham University), for their ability to form monolayers of polarized cells on two types of micro-carrier: Cytodex 3, i.e. dextran beads coated with pig skin gelatin, or polystyrene beads (Corning). As assessed by the cell morphology, the only successful combination was Caco-2/TC7 cells grown on Cytodex 3 beads. Caco-2/TC7 is a subclone of the parental Caco-2 human colon epithelial cell line, with a highly efficient ability to polarize and differentiate [13]. We verified that the resulting polarized Caco-2/TC7 monolayer grown on Cytodex 3 beads can be efficiently infected by an IAV and support multicycle growth of infectious progeny virions, therefore it is suitable to study any phase of IAV life cycle.

As a proof of concept, we documented the late stages of the viral life cycle using confocal and electron microscopy. IAVs have a segmented, single-stranded RNA genome of negative polarity. Fully infectious virions contain eight viral RNA (vRNA) segments that range in length from 0.9 to 2.3 kb, and together encode ten major and several auxiliary proteins. Each vRNA is bound by several copies of the viral nucleoprotein (NP) and one copy of the viral polymerase, to form viral ribonucleoproteins or vRNPs. After viral entry by endocytosis, incoming vRNPs are imported into the nucleus where transcription and replication of the viral genome takes place. Newly synthesized vRNPs exit the nucleus and are transported to the sites of viral assembly and budding at the plasma membrane [14]. Although there is evidence that the cellular small GTPase RAB11A, the endoplasmic reticulum, and to some extent microtubules and

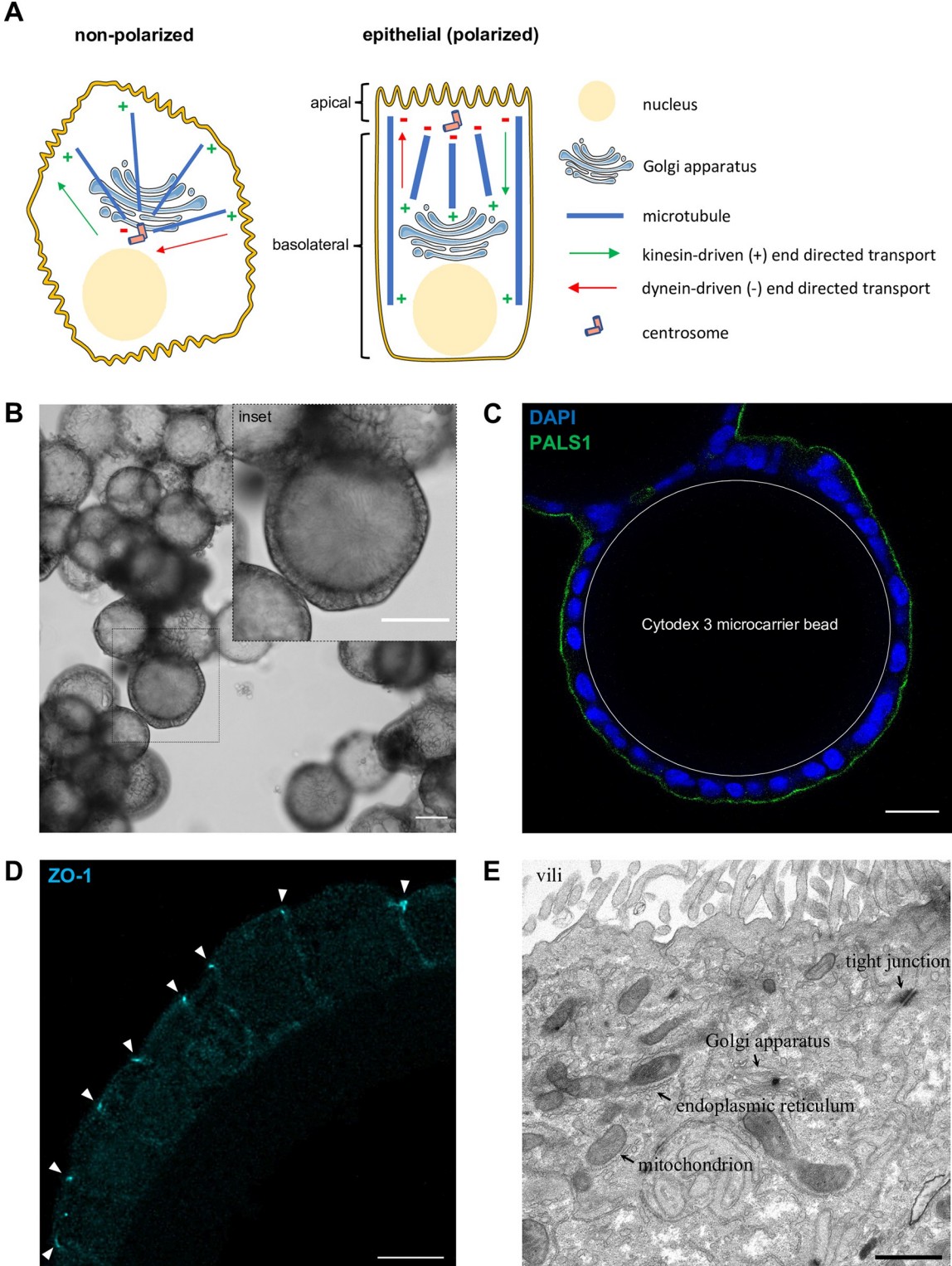

**Fig 1. Polarization of Caco-2/TC7 cells grown on Cytodex 3 microcarrier beads.** A. Schematic representation of the microtubule cytoskeleton organization in non-polarized versus polarized epithelial cells. In non-polarized cells the microtubules (in blue) grow from the centrosomes (pink cylinders) located near the nucleus and the Golgi apparatus towards the plasma membrane, whereas in polarized cells microtubules grow towards the basal pole. As a consequence, in non-polarized cells the molecular motors that move cargoes towards the plasma membrane are kinesins, the (+) end motors, whereas dynein, the (-) end motor, is moving cargoes towards the

apical membrane in polarized cells. B. Cellular morphology as observed with brightfield microscopy. A 10x objective was used (NA = 0.3, WD = 550mm). A bead cross-section is shown in the inset to illustrate the columnar morphology of Caco-2/TC7 cells at 14 days post-seeding on the beads. Scale bars: 100 µm. C. Confocal imaging upon immunostaining for the PALS1 apical membrane marker. A 40x objective (NA = 1.25, WD = 335 µm) was used. Blue and green colour: DAPI and PALS1 immunostaining, respectively. Step size: 0.3 µm, pixel size: 0.4 µm, speed: 400 Hz. Scale bar: 25 µm. D. Confocal imaging upon immunostaining for the ZO-1 marker of tight junctions labelled with ATTO647n. A 93x Leica glycerol objective (NA = 1.3, WD = 300 µm) was used. Blue colour: ZO-1 immunostaining. Step size: 0.18 µm, pixel size: 61 nm, speed: 600 Hz. Scale bar: 10 µm. E. Electron microscopy of an ultrathin section. Caco-2/TC7 cells grown on Cytodex 3 beads were fixed with glutaraldehyde, post-fixed with osmium tetroxide, dehydrated and resin-embedded for transmission electron microscopy. A representative ultrathin section going through the center of the Cytodex 3 beads is shown. Endoplasmic and Golgi apparatus membranes, as well as a mitochondrion and a tight junction, are indicated by arrows. Scale bar: 1 µm.

actin filaments are involved (for review [15]), the precise underlying mechanisms of vRNP transport remain largely obscure. As the cytoskeleton and secretion pathway are involved, it is particularly relevant to investigate these late stages of IAV life cycle in a polarized cell system.

Infected Caco-2/TC7 cells grown on Cytodex 3 beads were fixed in paraformaldehyde, co-stained with antibodies directed against RAB11A and the viral NP, the major component of vRNPs, and subsequently imaged using a Leica SP8 inverted confocal microscope equipped with 93x glycerol objective (NA = 1.3, WD = 300 µm) to provide an adequate depth of field and subcellular resolution. Alternatively, samples were fixed in glutaraldehyde, processed for negative stain and resin-embedded for transmission electron microscopy. Optical slices or ultrathin sections going through the center of the Cytodex 3 beads were selected, as they displayed the baso-apical axis of the polarized epithelial cells. By using this protocol, we were able to visualize along the baso-apical axis the colocalization of vRNPs and RAB11A in cytoplasmic structures, the accumulation of vRNPs at the apical membrane, and the presence of viral-induced arrays of endoplasmic reticulum (ER) structures oriented parallel to the apical membrane.

Our protocol provides a reproducible, low-tech, low-cost and fast approach to perform subcellular imaging of IAV-infected polarized human epithelial cells along the baso-lateral axis, with standard confocal and electron microscopes. A limitation is that, in our hands, it could not be extended to other human epithelial cells beyond the Caco-2/TC7 cells of intestinal origin, and these do not differentiate into a multi-cell type, mucus-secreting epithelium. However, compared to ALI cultures of human tracheo-bronchial cells, Caco-2/TC7 cells grown on Cytodex 3 beads present a number of advantages: in addition to being far more amenable to subcellular imaging, they provide a higher reproducibility, are easier to genetically engineer, require less expensive medium and consumables, and become polarized within two weeks instead of four to six weeks for ALI cultures. Importantly, Caco-2/TC7 cells grown on Cytodex 3 microcarrier beads recapitulate an essential feature of IAV natural tissue, which is polarity, and thereby allow to investigate the intracellular mechanisms of IAV replication in a context that is more relevant than non-polarized 2D cells cultures, notably regarding the trafficking of viral components in and out of the cell.

## Materials and methods

The protocol described in this manuscript is included for printing as **S1 Protocols**, and is published on protocols.io: dx.doi.org/10.17504/protocols.io.36wgq3pmklk5/v1.

## Expected results

Four days after seeding Caco-2/TC7 cells on approximately 1500 Cytodex 3 beads (175 µm in diameter), almost all beads are entirely covered with a confluent monolayer of cells. After filtering out of unattached cells and upon incubation of the beads for an additional period of 10–

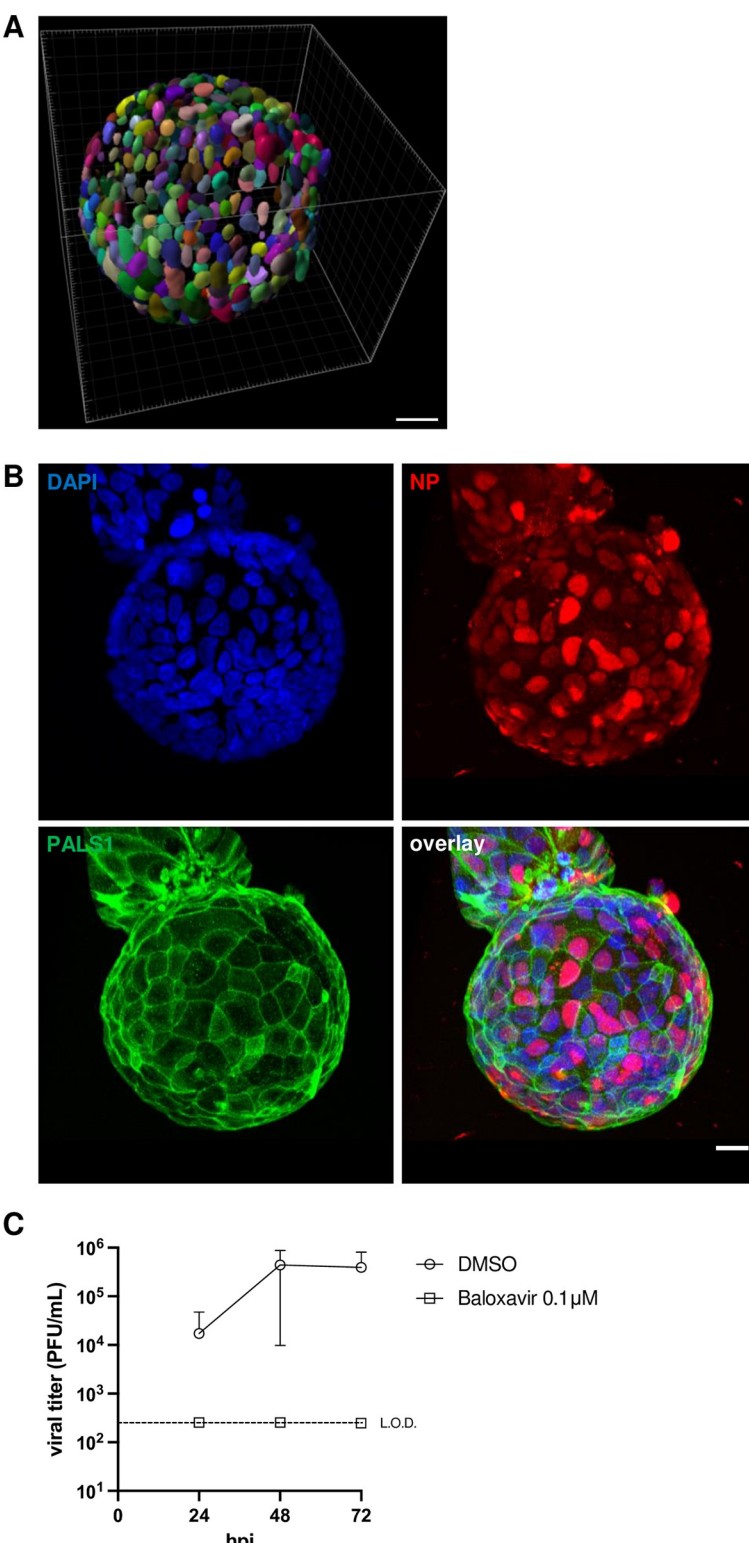

**Fig 2. Influenza A virus infection of polarized Caco-2/TC7 cells. A.** 3D-reconstructed z-stack of a Cytodex 3 microcarrier bead obtained by confocal imaging with a 40x objective (NA = 1.25, WD = 335 μm), segmented for DAPI-stained nuclei. The number of nuclei was determined using the Imaris software for cell segmentation. Scale bar: 40μm. **B.** 3D-reconstructed z-stack of half a Cytodex 3 microcarrier bead obtained by confocal imaging with a 40x objective (NA = 1.25, WD = 335 μm) upon immunostaining for the viral NP. Caco-2/TC7 cells grown on Cytodex 3

beads were infected at a MOI of 10 PFU/cell with the A/WSN/33 virus. At 8 hours post-infection (hpi) they were fixed with 4% paraformaldehyde and stained with a mix of antibodies specific for the viral NP or the cellular PALS1 protein, and DAPI for nuclear staining. Blue, red and green colour: DAPI, NP and PALS1 immunostaining, respectively. Step size: 0.3 μm, pixel size: 0.4 μm, speed: 400 Hz. Scale bar: 20 μm. **C.** Production of infectious IAV particles. Caco-2/TC7 cells grown on Cytodex 3 beads were infected at an estimated MOI of 0.001 PFU/cell with the A/WSN/33 virus, in the presence or absence of 0.1 μM of the viral inhibitor baloxavir. The supernatants were collected at 24, 48 and 72 hpi and titrated by plaque assay. The data are represented as the mean +/- SD of four independent experiments. The dashed line represents the limit of detection (L.O.D.) of 250 PFU/mL.

15 days, the cells acquire a columnar morphology (**Fig 1B**). Immunostaining for Polarity protein Associated with LIN Seven-1 (PALS1) and Zonula Occludens-1 (ZO-1) reveals the presence of an apical membrane domain (**Fig 1C**) and tight junctions (**Fig 1D**), respectively, at the distal pole with respect to the bead surface, clearly indicating the formation of a polarized monolayer. Microvilli, tight junctions and an apically located Golgi apparatus can be visualized by transmission electron microscopy (**Fig 1E**), further confirming that the Caco-2/TC7 cells are polarized after only 10–15 days of culture on Cytodex 3 beads. Polarized Caco-2/TC7 cells do not undergo cell division. Segmentation of a full-bead confocal image for DAPI-stained nuclei leads to an estimation of ~400 cells per bead (**Fig 2A**).

Unlike in polarized cysts, the apical membrane of Caco-2/TC7 cells polarized on Cytodex 3 beads is oriented towards the medium, which considerably facilitates infection assays. Upon infection with an IAV (A/WSN/33) at a high multiplicity of infection (MOI) of 10 PFU/cell, around 70% of the cells from the Caco-2/TC7 polarized monolayer are infected, as assessed by immunostaining for the viral NP protein and confocal imaging of full beads at 8 hours post-infection (hpi) (66.85% +/- 1.12% in three independent experiments, 2–5 beads examined in each experiment, approximately 80–100 cells per bead, 703 NP-positive cells out of 1053 in total) (**Fig 2B**). Upon infection at a low multiplicity of infection of 0.01 PFU/cell, infectious viral particles accumulate in the supernatant to titers in the range of $10^5$–$10^6$ PFU/mL. At 72 hpi, viral titers are at least 3-log lower in control beads treated with the specific IAV inhibitor baloxavir [16], which indicates that Caco-2/TC7 polarized monolayers support IAV multicycle growth and are therefore suitable to study any phase of IAV life cycle (**Fig 2C**). Seasonal influenza viruses with a monobasic cleavage site in the hemagglutinin (HA) were shown to undergo multiple replication sites in Caco-2 cells in the absence of exogenous trypsin, and there is evidence for the presence of an an intracellular trypsin-like protease that activates influenza HA in Caco-2 cells [17]. Therefore, the 3D cultured Caco-2/TC7 system is, in principle, not restricted to the WSN strain.

When non-polarized Caco-2/TC7 cells are infected with A/WSN/33 at an estimated MOI of 10 PFU/mL and stained for NP at 8 hpi, > 90% of them are NP-positive, however only few of them show vRNPs at the plasma membrane (**Fig 3A,** lower panels, white arrowhead, and **S1A Fig** for mock-infected control cells). Upon confocal imaging of half beads with a 40x objective (NA = 1.25, WD = 350 μm) and by selecting optical slices sections that go through the center of the Cytodex 3 beads, the accumulation of viral vRNPs can be visualized at the apical membrane of polarized Caco-2/TC7 cells (**Fig 3A,** upper panels, white arrowheads). At 8 hpi, the percentage of infected cells displaying NP signal at the apical/plasma membrane was 13.3% in polarized and 2.5% in non-polarized cells, respectively (p-value = 0.009) (**Fig 3B**), suggesting that distinct transport pathways (dynein- versus kinesin-driven) may be taking place, which may impact kinetics of the viral cycle.

By using confocal microscopy with a 93x objective (NA = 1.3, WD = 300 μm), it is possible to achieve subcellular resolution and to visualise NP expression in the nucleus of infected cells at 4 hpi, before it accumulates at the apical plasma membrane at 8 hpi and more markedly so

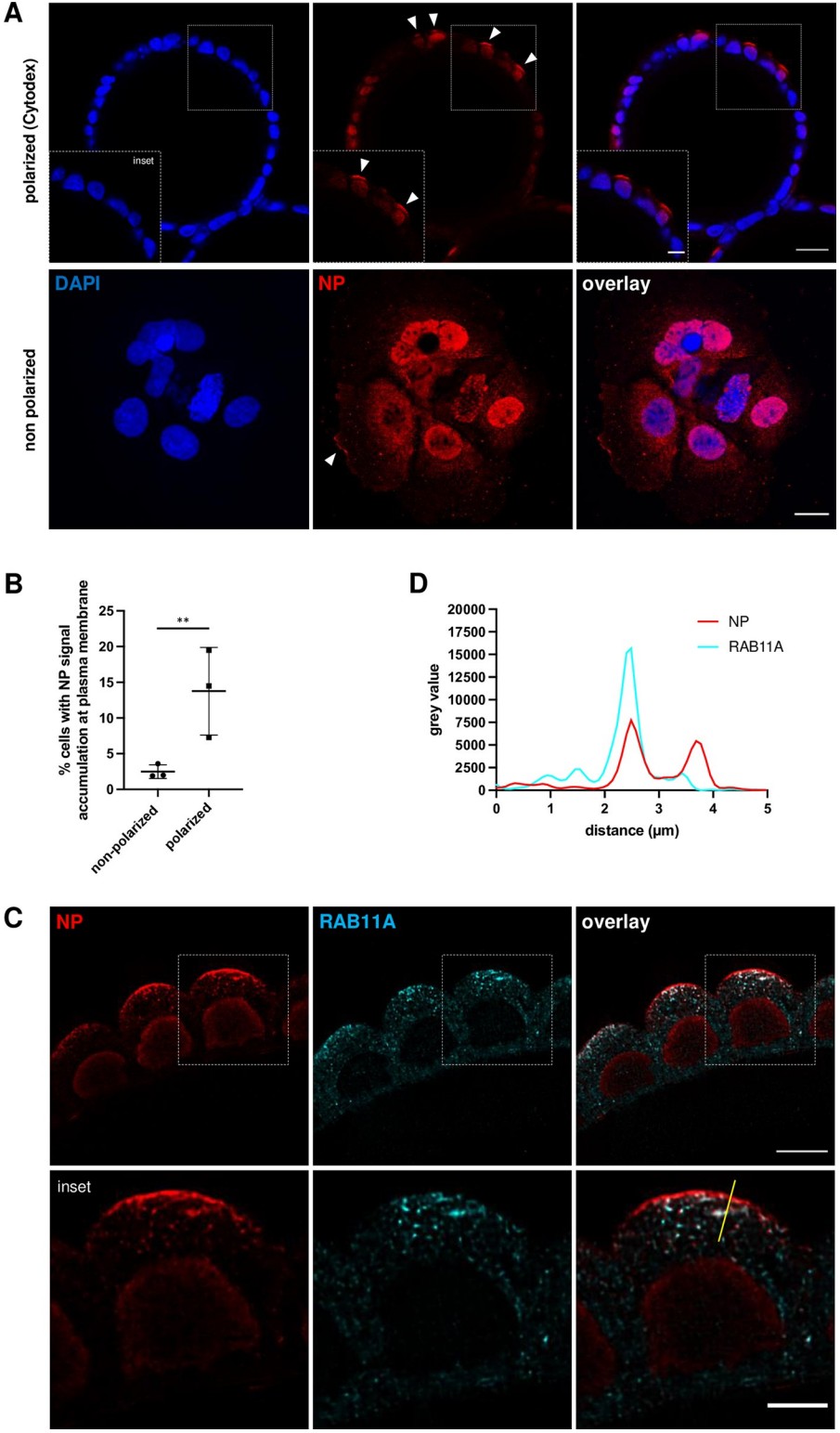

**Fig 3. Accumulation of influenza vRNPs at the apical membrane of polarized Caco-2/TC7 cells. A.** Caco-2/TC7 cells grown on Cytodex 3 beads (upper panels) or in standard 2D conditions (lower panels) were infected at an estimated MOI of 10 PFU/cell with the A/WSN/33 virus. At 8 hpi, they were fixed with 4% paraformaldehyde, stained with an antibody specific for the viral NP (red colour) and with DAPI for nuclear staining (blue color), and imaged using a confocal microscope with a 40x objective (NA = 1.25, WD = 335 μm). Upper panels: an optical slice going

through the center of a representative Cytodex 3 bead is shown. Step size: 0.3 μm, pixel size: 0.4 μm, speed: 400 Hz Scale bar: 25 μm. Inset scale bar: 10 μm. Lower panels: cells representative of the non-polarized state are shown. Step size: 0.3 μm, pixel size: 0.3 μm, speed: 400 Hz. Scale bar: 10 μm. White arrows indicate an accumulation of NP signal at the apical (upper panels) or plasma membrane (lower panels). **B.** Percentage of cells showing NP signal accumulation at the plasma/apical membrane in non-polarized versus polarized Caco-2/TC7 cells, at 8 hpi with the A/WSN/33 virus at a MOI of 10 PFU/cell. Three independent experiments were performed, in total 634 non-polarized and 703 polarized cells were randomly selected and inspected visually for NP signal accumulation at the plasma/apical membrane, as indicated by white arrows in panel A. Significance was tested with an unpaired t test after log10 transformation of the data, using GraphPad Prism. ** p-value = 0.009. **C.** Caco-2/TC7 cells grown on Cytodex 3 beads were infected at an estimated MOI of 10 PFU/cell with the A/WSN/33 virus. At 8 hpi they were fixed with 4% paraformaldehyde, co-stained with antibodies specific for the viral NP (red colour) and the cellular RAB11A protein (cyan colour), and imaged using a confocal microscope with a glycerol 93x objective (NA = 1.3, WD = 300 μm). Step size: 0.18 μm, pixel size: 85.7 nm, speed: 600 Hz. Images were deconvolved using the Huygens software. Scale bar: 10 μm. Inset scale bar: 5 μm. The yellow line corresponds to the intensity plot shown in panel D. **D.** Fluorescence intensity profile for NP (red colour) and RAB11A (cyan) along the yellow line drawn in Fig 3C (overlay inset). The X axis indicates the distance in μm from the proximal (close to the cell center) extremity of the line.

at 10 hpi (**S1B Fig**, arrowheads). At late time-points, the presence of viral NP-positive punctate structures within the cytoplasm of infected cells (**Fig 3C**, red colour) is observed, as well as their colocalization with the cellular RAB11A protein (**Fig 3C**, cyan colour). These double-stained structures resemble those reported previously in non-polarized cell systems (e.g. [18,19]) and most likely represent a cytosolic accumulation of vRNP transport vesicles. Interestingly, our imaging conditions also reveal that at 8 and 10 hpi, while the apical membrane is strongly positive for NP, it does not show any RAB11A signal (**Fig 3C** and a representative intensity plot in **Fig 3D**), suggesting that vRNPs and RAB11A are dissociated before the vRNPs reach the apical membrane.

Optical slices sections in the equatorial plane of the Cytodex 3 beads directly provide a view of polarized cells in the baso-lateral axis (**S2A and S2C Fig**). In contrast, when the cells are grown in 2D, direct access to a lateral view is not possible. Instead, an "orthogonal view" can be reconstructed from Z-stack acquisition of top view images in the x-y plane (**S2B and S2D Fig**). According to Abbe's diffraction theory, lateral resolution ($d_{xy} = \lambda/2NA$) is higher than the axial resolution ($d_{xz} = 2\lambda/(NA)^2$). In addition, Caco-2/TC7 cells grown on microcarriers adopt a cuboidal morphology whereas cells grown in 2D remain flat (**S2C Fig** compared to the orthogonal view in **S2D Fig**), which is also limiting spatial resolution (**S2E Fig** compared to **S2F Fig**). These physics features are the reason why 3D cell culture system on beads allows subcellular imaging along the baso-apical axis at a higher resolution compared to 2D cell culture systems.

Finally, electron microscopy of polarized Caco-2/TC7 cells on beads enables to observe IAV-induced remodeling of the endomembrane system. The images shown in **Fig 4** were obtained after the beads were fixed in glutaraldehyde, processed for negative staining, dehydrated and resin-embedded for electron microscopy. Ultrathin sections going through the center of the beads were selected and observed using a transmission electron microscope at 120 kV. In mock-infected cells, the ER structures are sparse and of no particular orientation with respect to the apical membrane (**Fig 4A**, left panel). In contrast, IAV-infected samples show arrays of elongated structures reminiscent of endoplasmic reticulum (ER) membranes oriented parallel to the apical membrane (**Fig 4A**, right panel). Long (> 1 μm) ER structures are more frequent in IAV-infected cells (p-value < 0.001) (**Fig 4B and 4C**). Interestingly, such arrays of ER structures parallel to each other and to the plasma membrane were not observed in IAV-infected 2D cell cultures, which underscores the interest to use polarized cell models that better recapitulate the natural target tissue of the IAV, and more largely respiratory viruses.

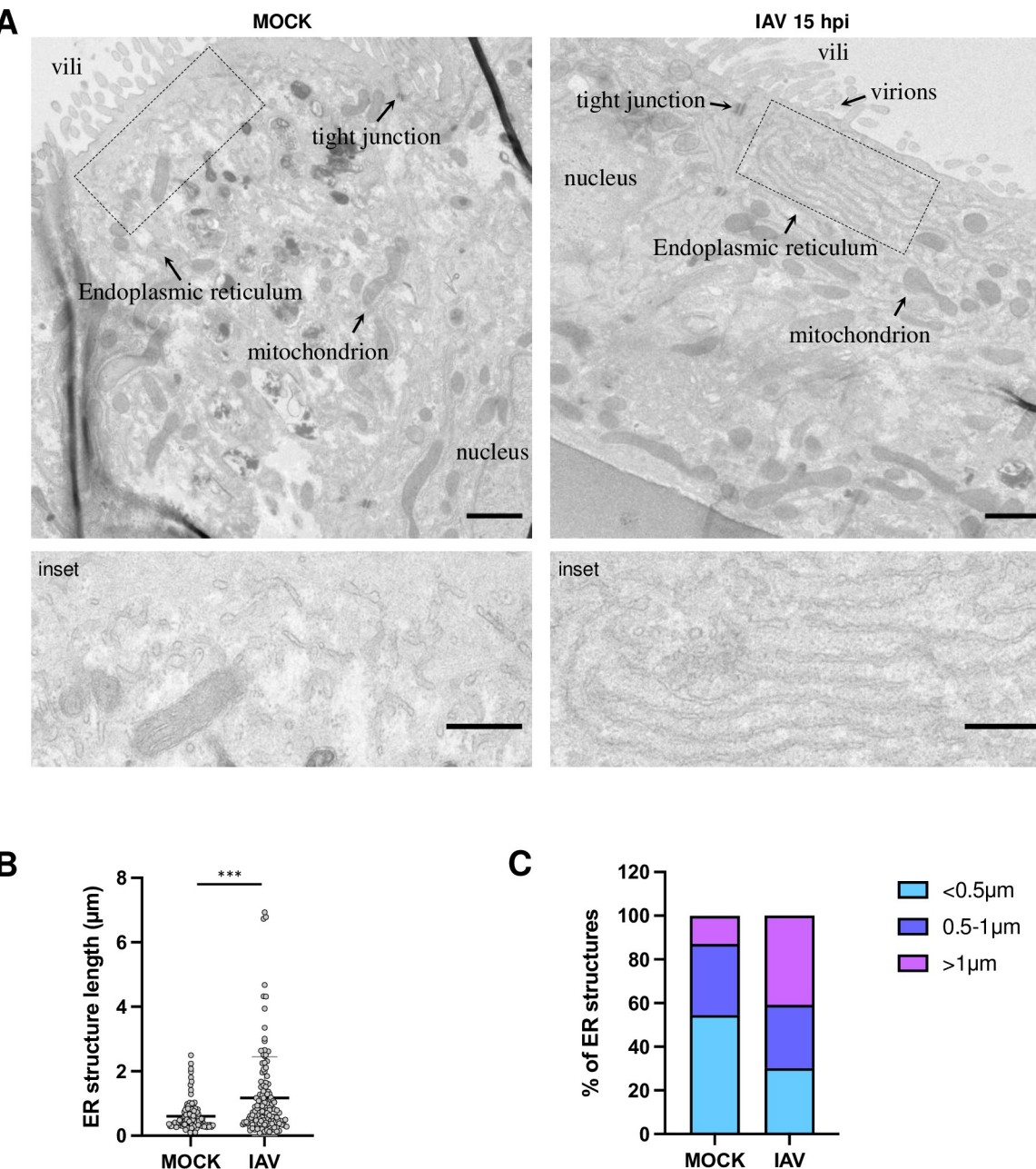

**Fig 4. Influenza virus-induced arrays of endoplasmic reticulum structures parallel to the apical membrane of polarized Caco-2/TC7 cells. A.** Caco-2/TC7 cells grown on Cytodex 3 beads were infected at an estimated MOI of 10 PFU/cell with the A/WSN/33 virus or mock-infected. At 15 hpi they were fixed with glutaraldehyde, post-fixed with osmium tetroxide, dehydrated and resin-embedded for transmission electron microscopy. Representative ultrathin sections going through the center of the Cytodex 3 beads are shown. The lower panels represent a higher magnification of the region defined by the dotted box within the upper panels. Endoplasmic and Golgi apparatus membranes, as well as mitochondrion and tight junctions, and, in the case of infected cells, extracellular virions, are indicated by arrows. Scale bar: 1μm. Scale bar for insets: 0.5 μm. **B-C.** Distribution of the length of endoplasmic reticulum (ER) structures as observed in IAV-infected versus mock-infected cells. Images in the.ser format were opened using the TIA reader plugin of Image J to determine the pixel size. Free-hand lines were drawn on apically located ER-like structures and their length was measured. Two independent experiments were performed. Five cells were used for each condition, 108 and 127 ER-like structures were measured for mock-treated and IAV-infected samples respectively. Significance was tested with an unpaired t test using GraphPad Prism. *** p-value<0.001.

## Supporting information

**S1 Fig. Accumulation of influenza vRNPs at the apical membrane of polarized Caco-2/ TC7 cells.**
(TIF)

**S2 Fig. Imaging the baso-lateral axis of Caco-2/TC7 cells grown on beads versus in a 2D system.**
(TIF)

**S1 Protocols.**
(PDF)

## Acknowledgments

We thank Véronique Carrière (Centre de Recherche Saint-Antoine, Paris, France) and Ian Sayers (Nottingham University, United Kingdom) for providing the Caco-2/TC7 cells and BMI-1 expressing bronchial epithelial cells, respectively. We thank David Hardy and Marie Anne Rameix-Welti (Institut Pasteur, Paris, France), and Vincent Fraisier and Chloé Guedj (Institut Curie, Paris, France) for helpful discussions. We thank the staff of NeurImag imaging core facility for their scientific expertise in data acquisitions.

## Author Contributions

**Conceptualization:** Jean-Baptiste Brault, Magda Cannata Serio, Nadia Naffakh.

**Funding acquisition:** Lydia Danglot, Adeline Mallet, Nadia Naffakh.

**Investigation:** Jean-Baptiste Brault, Catherine Thouvenot, Magda Cannata Serio, Sylvain Paisant, Julien Fernandes, David Gény, Lydia Danglot, Adeline Mallet.

**Supervision:** Nadia Naffakh.

**Writing – original draft:** Jean-Baptiste Brault, Nadia Naffakh.

**Writing – review & editing:** Jean-Baptiste Brault, Julien Fernandes, Lydia Danglot, Adeline Mallet, Nadia Naffakh.

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
