## [Decision Letter · Decision Letter 0]

27 Oct 2023

PONE-D-23-30615A polarized cell system amenable to subcellular resolution imaging of influenza virus infectionPLOS ONE

Dear Dr. Naffakh,

Thank you for submitting your manuscript to PLOS ONE. After careful consideration, we feel that it has merit but does not fully meet PLOS ONE’s publication criteria as it currently stands. Therefore, we invite you to submit a revised version of the manuscript that addresses the points raised during the review process.

Please, provide evidence about the superiority of your model over the 2-D cell culture model. As stated by one of the reviewers, a control with non-polarized cells should be included in Fig. 3. As suggested by the second reviewer, it is important to determine the MOI to be used; therefore, please, provide information on bead number and cell number over time. Also, provide more information on the step-by-step protocol in order to ease the investigators in using the method.

We look forward to receiving your revised manuscript.

Kind regards,

Boyan Grigorov

Academic Editor

PLOS ONE

Journal Requirements:

https://journals.plos.org/plosone/s/file?id=ba62/PLOSOne_formatting_sample_title_authors_affiliations.pd

2. We note you have not yet provided a protocols.io PDF version of your protocol and/or a protocols.io DOI. When you submit your revision, please provide a PDF version of your protocol as generated by protocols.io (the file will have the protocols.io logo in the upper right corner of the first page) as a Supporting Information file. The filename should be S1_file.pdf, and you should enter “S1 File” into the Description field. Any additional protocols should be numbered S2, S3, and so on. Please also follow the instructions for Supporting Information captions [https://journals.plos.org/plosone/s/supporting-information#loc-captions]. The title in the caption should read: “Step-by-step protocol, also available on protocols.io.”

Please assign your protocol a protocols.io DOI, if you have not already done so, and include the following line in the Materials and Methods section of your manuscript: “The protocol described in this peer-reviewed article is published on protocols.io (https://dx.doi.org/10.17504/protocols.io.[...]) and is included for printing purposes as S1 File.” You should also supply the DOI in the Protocols.io DOI field of the submission form when you submit your revision.

If you have not yet uploaded your protocol to protocols.io, you are invited to use the platform’s protocol entry service [https://www.protocols.io/we-enter-protocols] for doing so, at no charge. Through this service, the team at protocols.io will enter your protocol for you and format it in a way that takes advantage of the platform’s features. When submitting your protocol to the protocol entry service please include the customer code PLOS2022 in the Note field and indicate that your protocol is associated with a PLOS ONE Lab Protocol Submission. You should also include the title and manuscript number of your PLOS ONE submission.

4. Thank you for stating the following in the Acknowledgments Section of your manuscript: "We thank Véronique Carrière (Centre de Recherche Saint-Antoine, Paris, France) for providing the Caco-2/TC7 cells. We thank David Hardy and Marie Anne Rameix-Welti (Institut Pasteur, Paris, France), and Vincent Fraisier and Chloé Guedj (Institut Curie, Paris, France) for helpful discussions. This work was funded by a Human Frontiers Science Program (HFSP) grant held by NN (HFSP-RGP0040/2019), and a grant from the French National Research Agency (ANR) held jointly by NN and LD (FluNanotrack, ANR-21-CE11-0010-03). JBB was funded by the HFSP-RGP0040/2019 and the ANR-21-CE11-0010-03 grants. MCS was funded by the HFSP-RGP0040/2019 grant. 

The UtechS Photonic BioImaging (Imagopole), C2RT, Institut Pasteur is supported by the French National Research Agency (France BioImaging, ANR-10-INSB-04-01, ANR-10–INBS-04 and ANR-10-LABX-62-IBEID).

The NeurImag Imaging core facility at the Institute of Psychiatry and Neuroscience of Paris is a member of the national infrastructure France-BioImaging supported by the ANR (ANR-10-INBS-04). We thank the staff of NeurImag imaging core facility for their scientific expertise in data acquisitions and the Leducq foundation or funding the Leica SP8 Confocal/STED 3DX system."

Please remove any funding-related text from the manuscript and let us know how you would like to update your Funding Statement. Currently, your Funding Statement reads as follows: "NN : HFSP-RGP0040/2019

https://www.hfsp.org/

NN and LD : ANR-21-CE11-0010-03

https://anr.fr/

AM : ANR-10-INSB-04-01, ANR-10–INBS-04 and ANR-10-LABX-62-IBEID

https://anr.fr/

LD : ANR-10-INBS-04

https://anr.fr/

The funders did not and will not have a role in study design, data collection and analysis, decision to publish, or preparation of the manuscript."

Reviewers' comments:

Reviewer's Responses to Questions

**Comments to the Author**

1. Does the manuscript report a protocol which is of utility to the research community and adds value to the published literature?

Reviewer #1: Yes

Reviewer #2: Yes

2. Has the protocol been described in sufficient detail?

To answer this question, please click the link to protocols.io in the Materials and Methods section of the manuscript (if a link has been provided) or consult the step-by-step protocol in the Supporting Information files.

The step-by-step protocol should contain sufficient detail for another researcher to be able to reproduce all experiments and analyses.

Reviewer #1: Yes

Reviewer #2: Partly

3. Does the protocol describe a validated method?

Reviewer #1: No

Reviewer #2: Yes

4. If the manuscript contains new data, have the authors made this data fully available?

Reviewer #1: Yes

Reviewer #2: Yes

**5. Is the article presented in an intelligible fashion and written in standard English?**

Reviewer #1: Yes

Reviewer #2: Yes

6. Review Comments to the Author

Reviewer #1: the authors describe a method of 3D culture for Caco2 cells on a Cytodex bead carrier. They convincingly show that cells polarise on these beads and suggest that this model could be used to study process in influenza A virus infected cells, which require e.g. the specific organisation of the cytoskeleton that is found in polartized but not in non polarised cells.

Specific points:

This technique is not new and has been used to study enterovirus infection (PMID: 27303677).

The intestinal origins of the cells is not ideal considering that the authors make a rather strong point about the relevance of this new model.

The manuscript lacks a functional proof of superiority over 2D culture of Caco2 cells. While the NP localisation phenotype seems clear, it remains to be shown if this affects the replication of the virus? One could imagine that polymerase transport between cytoplasm and nucleus might be quite different in polimerized vs non-polimerized cells.

Missing controls: Fig 3 would improve with non-infected controls. imaging of non-polarised cells should be included.

In Fig 4: if the scale bar is 1um the indicated vision is quite small (estimated <80nm) for a influenza A virus particle

Reviewer #2: In this manuscript, Brault et al. describe a detailed and comprehensive protocol to generate polarized cell monolayers on microbeads to study influenza virus replication. Intracellular organization of these polarized cells resemble the intracellular organization found in the natural target cells of the virus, the respiratory epithelial cells, whereas the cell lines classically used for flu infections are cultured in 2D, under non-polarizing conditions. This is an important issue as molecular motors, for example, seem to act in opposite directions between polarized and non-polarized cells. Thus, it appears essential to study events such as intracellular transport of viral elements and the cellular factors involved, in relevant, polarized cellular systems.

The authors used a specific cell line (Caco-2/TC7 cells, human) and a specific type of beads (Cytodex-3) to produce a polarized cell monolayer, allowing visualization of the baso-apical axis of the polarized cells and subcellular resolution imaging.

They demonstrated the ability of the polarized cells to replicate influenza A virus (WSN strain) and illustrated this 3D cell culture system with some spectacular microscopy images. This system allowed some interesting observations such as the absence of colocalization between NP (likely vRNPs) and the cellular small GTPase RAB11A at 8h post-infection at the apical membrane of polarized cells shown by fluorescence intensity profile (but both colocalized in non-polarized cells), whereas the two proteins colocalized in the cytoplasm, in the two systems. Comparison between polarized and non-polarized cells points towards different vRNPs transport pathways, something that should be considered when studying influenza virus life cycle. Moreover, the presence of viral-induced rearrangement of the endoplasmic reticulum was observed by electron microscopy in polarized cells only.

Having this protocol published by PlosOne will benefit to the study of influenza viral cycle, but will also be useful more broadly to the study the replication of any intracellular pathogen able to infect Caco2/TC7 cells (not only viruses). A few remarks are listed below in order to improve the manuscript.

In the “influenza A virus infection” section, the determination of the bead concentration / number, after the successive washing steps and before infection, is not mentioned. This is important to determine the MOI to use. Could it be useful / important to count the number of beads?

Along these lines, p. 5, it is mentioned that four days after cell seeding on the beads, around 600 cells are present per bead. Is this number the same 10-15 days later, when the cells are infected? There is no more cell division between day 4 and 10-15? Again, this is relevant regarding MOI calculations and the authors should specify this.

In this study, WSN was used. This strain doesn’t need the addition of TPCK-treated trypsin in the culture supernatant to perform multirounds of replication, but it is an exception in that regard. What if another strain than WSN, meaning needing TPCK-trypsin in the media for production of infectious particles and multirounds of replication, is used? Do the Caco-2/TC7 coated beads support a concentration of TPCK trypsin compatible with more classical influenza strains? If not, this system will still be useful for single-round infection studies, but this should be specified in the text.

Figure 3B : what about later time points? Does the accumulation of NP at the plasma membrane increase over time?

Additional, minor comments.

Abstract, first line : depend -> depends

p.4 Caco-2/TC7 cells grown on Cytodex 3 beads were fixed… -> Infected Caco-2/TC7 cells grown on…

p.4 Alternatively, they were fixed… -> Alternatively, samples were fixed…

p.5-6 “Indeed, upon infection with an IAV (A/WSN/33) at a high multiplicity of infection of 10 PFU/cell, around 70% of the cells from the Caco-2/TC7 polarized monolayer are infected as assessed by immunostaining for the viral NP protein and confocal imaging of full beads (66.85% +/- 1.12% in three independent experiments, 2-5 beads examined in each experiment, approximately 80-100 cells per bead, 703 NP-positive cells out of 1053 in total) « -> please, add the time-point used (8 hpi). Is this percentage similar in non-polarized cells?

p. 6 … it is possible to reach subcellular resolution, and observe the accumulation… -> and to observe…

p. 9 Fig 2 legend, B. …72 hpi and titrated using a plaque assay -> titrated by plaque assay

p. 9 Fig 3 legend, A. … At 8 hpi they were fixed… -> At 8 hpi, they were fixed…

p. 10 Fig 3 legend, A. …defined by the dotted line -> define by the dotted box

p. 10 Fig 3 legend, B-C. …Image J to determine the pixel size. Plugin. Free-hand lines were drawn… -> remove “Plugin.”

Step-by-step protocol

Please add more technical details and references for the readers :

1.1: can you specify the Gelrite concentration, solution used for preparation and supplier?

1.2: … in a 35 mm dish or a well or a 6-well plate -> … in a 35 mm dish or a well of a 6-well plate

1.4: specify trypsin reference and supplier

1.7: specify Cytodex 3 reference and supplier

3.3: … add 2 mL of fresh D10 on the top… -> … add 2 mL of fresh D10 medium on the top…

3.4: … D10 -> D10 medium

7. Start with “Let the beads sediment and remove…”

7. …(estimated number of cells per beads:). Please, add 600 if correct.

12a / 12.1: specify the composition of the blocking buffer used?

12a / 12.2: remove “Remove the blocking buffer and add 250 μL of the primary antibody diluted in the immuno-staining buffer.” As the sentence is there twice.

12a / 12.2: specify the composition of the immune-staining buffer used?

12a / 12.6: specify the supplier and the reference for the Vectashield mounting medium?

13a / 13.3: can you specify the references of the primary and secondary antibodies used?

11b / 11.2: PEM definition

13b: remove capital E from ethanol

13b / 13.4: can you give more details on the Epoxy resin preparation?

13b / 13.6: … slowly with with back-and-forth… -> remove 1 with

13b / 13.7 Remove 150 uL of Epoxy resin

14b / 14.12: toludine blue concentration?

15b / 15.5: lead citrate concentration?

7. PLOS authors have the option to publish the peer review history of their article (what does this mean?). If published, this will include your full peer review and any attached files.

Reviewer #1: No

Reviewer #2: No

---

## [Author Response · Author response to Decision Letter 0]

27 Dec 2023

We would like to thank the reviewers for their constructive comments. In the revised version of the manuscript, additional data have been included and the text has been modified in order to address their concerns and suggestions. 

A revised version of the manuscript with highlighted changes is provided. 

Below is a point-by-point response to each of the reviewers’ comments.

Reviewer #1: the authors describe a method of 3D culture for Caco2 cells on a Cytodex bead carrier. They convincingly show that cells polarise on these beads and suggest that this model could be used to study process in influenza A virus infected cells, which require e.g. the specific organisation of the cytoskeleton that is found in polartized but not in non polarised cells.

Specific points:

This technique is not new and has been used to study enterovirus infection (PMID: 27303677).

We acknowledge that the 3D culture of Caco2 cells on Cytodex beads is not a novel technique per se and has previously been used to study bacterial and viral infections. In the present manuscript however, our aim is to demonstrate and describe how this 3D culture system can be used to achieve subcellular imaging of influenza virus-infected cells with a high resolution along the baso-lateral axis. To our best knowledge, that specific use has not been described previously. 

In the initial version of our manuscript we cited the article by Jakob et al. (PMID 26847791), which describes the 3D static culture of Caco2 cells on Cytodex beads which we used. 

In the revised version (page 3 line 34, page 4 line 1) we included additional references to Drummond et al 2015 (PMID: 27303677), as suggested by the reviewer, and Papafragkou et al 2013 (PMID: 23755105), who used 3D rotating culture of Caco2 cells on Cytodex beads to study enterovirus and calicivirus infection, respectively. 

The intestinal origins of the cells is not ideal considering that the authors make a rather strong point about the relevance of this new model.

We agree that cells of respiratory origin would have been most relevant. 

This is precisely the reason why, as stated in our manuscript (page 4 lines 1-6 of the initial and the revised version), we tested, along with Caco-2 cells, several human respiratory cell lines (A549, Calu-3, NCI-H292, RPMI-1, 16HBE14), as well as quasi-primary, BMI-1-expressing bronchial epithelial cells derived from two distinct patients, for their ability to form monolayers of polarized cells on two types of micro-carriers. 

The only cells that were able to form monolayers of polarized cells were the Caco-2/TC7 cells, when grown on Cytodex 3 beads. We believe that it is useful to let the readers know that our attempts to adapt this 3D cell culture system to several common respiratory cell lines were unsuccesful, in order to reduce duplication of effort and stimulate the investigation of alternative systems. 

In the revised version we have added the terms “of intestinal origin” when we discuss the limitations of this model (page 5 line 4 of the revised version) : “A limitation is that, in our hands, it could not be extended to other human epithelial cells beyond the Caco-2/TC7 cells of intestinal origin, and these do not differentiate into a multi-cell type, mucus-secreting epithelium”. 

Nevertheless, we believe that the major asset of our model is that it recapitulates an essential feature of IAV natural tissue, i.e. the establishment and maintenance of apical polarity. Major features of apical polarity are shared in intestinal and respiratory epithelial cells, notably the accumulation of microtubule minus-ends at the apical surface (Toya et al, PMID 26715742; Robinson et al 2020, PMID 32482850). Therefore, Caco-2 cells grown on beads represent a valuable tool to investigate the intracellular mechanisms of IAV replication, notably regarding the trafficking of viral components in and out of the cell. 

The manuscript lacks a functional proof of superiority over 2D culture of Caco2 cells. 

The major advantage of the 3D cell culture on beads is that imaging of the equatorial plane directly provides a lateral view of polarized cells, which allows high-resolution imaging along the baso-lateral axis (z-axis). In contrast, with classical 2D cell culture systems, direct imaging in the z-axis is not possible. Lateral views of the cells can only be reconstructed from a series of orthogonal images taken from the top, in the x-y plane, which results in lower resolution. 

To provide evidence about the superiority of the 3D culture over the 2D culture of Caco-2/TC7 cells in terms of imaging along the baso-lateral axis, we have included confocal microscopy images of IAV-infected Caco-2/TC7 cells cultured in standard 2D conditions. Thereby, we illustrate the much lower Z axis resolution achieved compared to the 3D model and we strengthen our claim that the 3D model, unlike the 2D model, allows imaging along the baso-lateral axis of polarized cells at a subcellular resolution. 

In the revised version of the manuscript, these new data are presented in the Supplementary Fig S2, and are discussed page 7 lines 4-13.

While the NP localisation phenotype seems clear, it remains to be shown if this affects the replication of the virus? One could imagine that polymerase transport between cytoplasm and nucleus might be quite different in polimerized vs non-polimerized cells.

Although we discuss the fact that being able to achieve subcellular resolution is a strong asset to study the biology of pathogens that infect polarized cells, we feel that including functional studies for an extensive comparison of the 2D versus 3D model is beyond the scope of this Lab Protocol article.

Missing controls: Fig 3 would improve with non-infected controls. imaging of non-polarised cells should be included.

Imaging of non-polarized cells is already included in Fig. 3A of the initial version of our manuscript.

To complement Fig. 3, we included 

- Confocal microscopy images of non-infected cells in the Supplementary Fig S1A, in order to demonstrate the specificity of the NP antibody. 

- Confocal microscopy images of non-polarized cells in Supp Fig S2, to provide evidence about the superiority of the 3D culture over the 2D culture of Caco-2 cells for imaging along the baso-lateral axis at a subcellular resolution (detailed in the response to the preceding comment)

The text and figure legends have been modified accordingly, page 6 lines 16-18 and page 7 lines 4-13). 

In Fig 4: if the scale bar is 1um the indicated vision is quite small (estimated <80nm) for a influenza A virus particle

We double-checked that the scale bar is correct in Fig 4. 

Influenza virions are reported to be 80-120 nm in diameter, and in a recently published article (Vale-Costa et al 2023, PMID 37983294, Fig 2D), cryo-electron tomography images of infected A549 cells show virions in the same size range as in Fig 4 of our manuscript. 

Reviewer #2: In this manuscript, Brault et al. describe a detailed and comprehensive protocol to generate polarized cell monolayers on microbeads to study influenza virus replication. Intracellular organization of these polarized cells resemble the intracellular organization found in the natural target cells of the virus, the respiratory epithelial cells, whereas the cell lines classically used for flu infections are cultured in 2D, under non-polarizing conditions. This is an important issue as molecular motors, for example, seem to act in opposite directions between polarized and non-polarized cells. Thus, it appears essential to study events such as intracellular transport of viral elements and the cellular factors involved, in relevant, polarized cellular systems.

The authors used a specific cell line (Caco-2/TC7 cells, human) and a specific type of beads (Cytodex-3) to produce a polarized cell monolayer, allowing visualization of the baso-apical axis of the polarized cells and subcellular resolution imaging.

They demonstrated the ability of the polarized cells to replicate influenza A virus (WSN strain) and illustrated this 3D cell culture system with some spectacular microscopy images. This system allowed some interesting observations such as the absence of colocalization between NP (likely vRNPs) and the cellular small GTPase RAB11A at 8h post-infection at the apical membrane of polarized cells shown by fluorescence intensity profile (but both colocalized in non-polarized cells), whereas the two proteins colocalized in the cytoplasm, in the two systems. Comparison between polarized and non-polarized cells points towards different vRNPs transport pathways, something that should be considered when studying influenza virus life cycle. Moreover, the presence of viral-induced rearrangement of the endoplasmic reticulum was observed by electron microscopy in polarized cells only.

Having this protocol published by PlosOne will benefit to the study of influenza viral cycle, but will also be useful more broadly to the study the replication of any intracellular pathogen able to infect Caco2/TC7 cells (not only viruses). A few remarks are listed below in order to improve the manuscript.

In the “influenza A virus infection” section, the determination of the bead concentration / number, after the successive washing steps and before infection, is not mentioned. This is important to determine the MOI to use. Could it be useful / important to count the number of beads?

Along these lines, p. 5, it is mentioned that four days after cell seeding on the beads, around 600 cells are present per bead. Is this number the same 10-15 days later, when the cells are infected? There is no more cell division between day 4 and 10-15? Again, this is relevant regarding MOI calculations and the authors should specify this. 

Additional information is provided in the revised version of the protocol, to facilitate the determination of infection conditions at a given MOI. 

The Cytodex 3 beads (3 x 106 beads per g according to the supplier’s indication) are initially weighted and resuspended at a concentration of 1g per 100 mL of PBS, which corresponds to 1500 beads per 50 µL (step 1.7 of the protocol). 

Once Caco-2/TC7 cells are polarized on the Cytodex 3 beads, they no longer divide. We counted the number of polarized cells per bead upon segmentation of a full-bead confocal image for DAPI-stained nuclei, and found ~400 cells per bead. In the revised manuscript this method is illustrated in an additional Figure (Fig 2A) and commented page 5 lines 29-30. 

A theoretical calculation, based on the average surface area of a bead given the diameter of ~175 µm (~ 96,200 µm2) and estimated surface area of a Caco-2 cell based on brifghfield imaging of bead cross-sections (~ 225 µm2), leads to the same estimation of ~400 cells per bead. 

This estimated number of cells per sample (1,500 beads x 400 cells per bead = 600000 cells) serves as a basis to calculate the amount of infectious viral particles (PFU) required to achieve a given MOI, e.g. 6 x 106 PFU, or 250 µL of a viral suspension at titer of ~2,4 x107 PFU/mL, will be required to achieve a MOI of 10 (step 7 of the protocol). 

In this study, WSN was used. This strain doesn’t need the addition of TPCK-treated trypsin in the culture supernatant to perform multirounds of replication, but it is an exception in that regard. What if another strain than WSN, meaning needing TPCK-trypsin in the media for production of infectious particles and multirounds of replication, is used ? Do the Caco-2/TC7 coated beads support a concentration of TPCK trypsin compatible with more classical influenza strains? If not, this system will still be useful for single-round infection studies, but this should be specified in the text.

We thank the reviewer for this comment. 

Seasonal influenza viruses with a monobasic cleavage site in the HA were shown to undergo multiple replication sites in Caco-2 cells in the absence of exogenous trypsin, and there is evidence for the presence of an an intracellular trypsin-like protease that activates influenza HA in Caco-2 cells (Zhirnov et al, 2003, PMID : 12951033). Therefore, the 3D cultured Caco-2/TC7 system is, in principle, not restricted to the WSN strain. 

This statement was included in the revised version of the manuscrit (page 6 lines 11-16). 

Figure 3B : what about later time points? Does the accumulation of NP at the plasma membrane increase over time?

The time point shown in Figure 3B (renumbered Figure 4A in the revised version) is 8 hours post-infection. 

In the revised version, we included two additional time points : at 4 hpi there is no detectable NP at the plasma membrane, whereas at 10 hours post-infection the accumulation of NP at the plasma membrane is more pronounced and observed in a higher proportion of cells. 

These additional data are presented in the Supplementary Fig S1B, and are mentioned page 6 lines 27-30 of the revised version. 

Additional, minor comments.

We thank the reviewer for his/her very careful reading of the protocol and insightful suggestions. They have all been taken into account in the revised version of our manuscript and the associated step-by-step protocol.

Abstract, first line : depend -> depends

p.4 Caco-2/TC7 cells grown on Cytodex 3 beads were fixed… -> Infected Caco-2/TC7 cells grown on…

p.4 Alternatively, they were fixed… -> Alternatively, samples were fixed…

p.5-6 “Indeed, upon infection with an IAV (A/WSN/33) at a high multiplicity of infection of 10 PFU/cell, around 70% of the cells from the Caco-2/TC7 polarized monolayer are infected as assessed by immunostaining for the viral NP protein and confocal imaging of full beads (66.85% +/- 1.12% in three independent experiments, 2-5 beads examined in each experiment, approximately 80-100 cells per bead, 703 NP-positive cells out of 1053 in total) « -> please, add the time-point used (8 hpi). Is this percentage similar in non-polarized cells?

The percentage in non-polarized cells is around 90% of infected cells under these conditions. This information was added page 6 line 18 of the revised version. 

p. 6 … it is possible to reach subcellular resolution, and observe the accumulation… -> and to observe…

p. 9 Fig 2 legend, B. …72 hpi and titrated using a plaque assay -> titrated by plaque assay

p. 9 Fig 3 legend, A. … At 8 hpi they were fixed… -> At 8 hpi, they were fixed…

p. 10 Fig 3 legend, A. …defined by the dotted line -> define by the dotted box

p. 10 Fig 3 legend, B-C. …Image J to determine the pixel size. Plugin. Free-hand lines were drawn… -> remove “Plugin.”

Step-by-step protocol

Please add more technical details and references for the readers :

1.1: can you specify the Gelrite concentration, solution used for preparation and supplier?

1.2: … in a 35 mm dish or a well or a 6-well plate -> … in a 35 mm dish or a well of a 6-well plate

1.4: specify trypsin reference and supplier

1.7: specify Cytodex 3 reference and supplier

3.3: … add 2 mL of fresh D10 on the top… -> … add 2 mL of fresh D10 medium on the top…

3.4: … D10 -> D10 medium

7. Start with “Let the beads sediment and remove…”

7. …(estimated number of cells per beads:). Please, add 600 if correct.

12a / 12.1: specify the composition of the blocking buffer used?

12a / 12.2: remove “Remove the blocking buffer and add 250 μL of the primary antibody diluted in the immuno-staining buffer.” As the sentence is there twice.

12a / 12.2: specify the composition of the immune-staining buffer used?

12a / 12.6: specify the supplier and the reference for the Vectashield mounting medium?

13a / 13.3: can you specify the references of the primary and secondary antibodies used?

11b / 11.2: PEM definition

13b: remove capital E from ethanol

13b / 13.4: can you give more details on the Epoxy resin preparation?

13b / 13.6: … slowly with with back-and-forth… -> remove 1 with

13b / 13.7 Remove 150 uL of Epoxy resin

14b / 14.12: toludine blue concentration?

15b / 15.5: lead citrate concentration?

---

## [Editor Report · Decision Letter 1]

29 Dec 2023

A polarized cell system amenable to subcellular resolution imaging of influenza virus infection

PONE-D-23-30615R1

Dear Dr. Naffakh,

We’re pleased to inform you that your manuscript has been judged scientifically suitable for publication and will be formally accepted for publication once it meets all outstanding technical requirements.

Kind regards,

Boyan Grigorov

Academic Editor

PLOS ONE
---

## [Editor Report · Acceptance letter]

17 Jan 2024

PONE-D-23-30615R1 

PLOS ONE

Dear Dr. Naffakh, 

I'm pleased to inform you that your manuscript has been deemed suitable for publication in PLOS ONE. Congratulations! Your manuscript is now being handed over to our production team.

Kind regards, 

on behalf of

Dr. Boyan Grigorov 

Academic Editor

PLOS ONE